# Psychometric Validation of the Cardiff Wound Impact Schedule Questionnaire in a Spanish Population with Diabetic Foot Ulcer

**DOI:** 10.3390/jcm10174023

**Published:** 2021-09-06

**Authors:** Minerva Granado-Casas, Dolores Martinez-Gonzalez, Montserrat Martínez-Alonso, Montserrat Dòria, Nuria Alcubierre, Joan Valls, Josep Julve, José Verdú-Soriano, Didac Mauricio

**Affiliations:** 1Lleida Institute for Biomedical Research Dr. Pifarré Foundation IRBLleida, University of Lleida, Av. Alcalde Rovira Roure 80, 25198 Lleida, Spain; minervagranado@outlook.com (M.G.-C.); lmartinezlleida@gmail.com (D.M.-G.); montserratdoria@gmail.com (M.D.); nurialcubierre@gmail.com (N.A.); 2Department of Endocrinology & Nutrition, Hospital de la Santa Creu i Sant Pau, Sant Quintí 89, 08041 Barcelona, Spain; JJulve@santpau.cat; 3Centre for Biomedical Research on Diabetes and Associated Metabolic Diseases (CIBERDEM), Instituto de Salud Carlos III, 08907 Barcelona, Spain; 4Systems Biology and Statistical Methods for Biomedical Research, IRBLleida, University of Lleida, Av. Alcalde Rovira Roure 80, 25198 Lleida, Spain; mmartinez@irblleida.cat (M.M.-A.); jvalls@irblleida.cat (J.V.); 5Diabetic Foot Unit, University Hospital Arnau de Vilanova, Rovira Roure 80, 25198 Lleida, Spain; 6Department of Community Nursing, Preventive Medicine & Public Health and History of Science, University of Alicante, Carretera de Sant Vicent del Raspeig s/n, 03080 Alicante, Spain; pepe.verdu@ua.es; 7Grupo Nacional de Estudio y Asesoramiento de Úlceras por Presión (GNEAUPP) Steering Committee, 26004 Logroño, Spain; 8Department of Medicine, Faculty of Medicine, University of Vic & Central University of Catalonia, 08500 Vic, Spain

**Keywords:** diabetic foot ulcer, type 2 diabetes, quality of life, psychometric validation, reliability, validity

## Abstract

Diabetic foot ulcers (DFU) negatively affect the quality of life (QoL) of people with diabetes. The Cardiff Wound Impact Schedule (CWIS) questionnaire has been designed to measure the QoL of people with chronic foot wounds. However, no studies have been specifically designed to validate this instrument in a Spanish population. In this prospective study, a total of 141 subjects with DFU were recruited. DFU was determined by performing physical examinations. Medical records were exhaustively reviewed to collect clinical variables. The CWIS was transculturally adapted by a group of experts and a group of patients with DFU. The SF-36 and EQ-5D generic instruments were used as reference tools. The questionnaires were administered at 7 days and 4, 12, and 26 weeks after the baseline assessment by personal interview with each of the study subjects. The psychometric properties of the instrument were assessed using statistical methods. The content validity had an average of 3.63 (90.7% of the maximum score of 4). The internal consistency of the CWIS subscales had a standardized Cronbach’s alpha range from 0.715 to 0.797. The reproducibility was moderate with an intraclass correlation coefficient (ICC) range from 0.606 to 0.868. Significant correlations between CWIS domains and SF-36 and EQ-5D subscales were observed, demonstrating a good criterion validity of the CWIS questionnaire (*p* < 0.001). However, the construct validity of the CWIS was not validated with a comparative fit index (CFI) of 0.69, a root mean square error of approximation (RMSEA) of 0.09, and a standardized root mean square residual (SRMR) of 0.10. The sensitivity to changes over time was optimal in the three domains (i.e., social life, well-being, and physical symptoms) (*p* < 0.001). In conclusion, the Spanish version of the CWIS shows acceptable psychometric properties to assess the QoL of subjects with DFU, except for its construct validity.

## 1. Introduction

Diabetic foot ulcer (DFU) is an important complication of diabetes, with an incidence rate of 1–4% and a lifetime risk of 15–25% [1,2]. This condition is defined as an ulceration of the foot associated with diabetic neuropathy which show any grade of ischemia and infection [3]. Moreover, this diabetic complication is often associated with other serious complications such as osteomyelitis and lower limb amputation [2]. The five-year mortality rate among patients with DFU is around 40%, with a 2.5 times higher risk of mortality in comparison with patients without DFU [4]. The pathogenesis of DFU involves multiple factors, such as peripheral neuropathy, artery disease, traumas and foot deformities, as well as abnormal joints [3]. Additionally, a recent meta-analysis about the global epidemiology of DFU demonstrated that subjects with DFU were older, had lower body mass index, longer diabetes duration, and showed higher frequency of hypertension, diabetic retinopathy, and smoking habit in comparison with those without DFU [5]. Additionally, DFU negatively impacts the health-related quality of life (HRQoL) of the affected patients, especially those with unhealed ulcers [6,7,8,9,10,11].

HRQoL is a patient-reported outcome (PRO) that takes into account the presence of biological or physiological dysfunction, symptoms and functional impairment [12]. Moreover, quality of life (QoL) is a multidimensional, subjective, and dynamic measure of the physical, psychological, and social aspects of daily life [12]. Patients with DFU report pain, and limited daily and social activities that worsen their QoL [2]. Moreover, a poorer QoL is associated with several clinical factors, such as pain, fatigue, wound infection, restricted mobility, and social isolation [13]. Furthermore, QoL can be negatively influenced by other factors such as the frequency of attending clinic visits and hospitalizations, and the presence of disturbed daily life activities [13]. The importance of measuring a PRO can be focused on providing valuable information about the effectiveness of a treatment or intervention care [4]. In addition, it can help us to understand how DFUs impact on patients’ QoL, with the aim of helping to improve their health care [4].

The Cardiff Wound Impact Schedule (CWIS) questionnaire was designed and validated to assess the impact of chronic wounds on the QoL of patients [14,15]. Although this is not a specific questionnaire for DFU, it is able to discriminate between healed and unhealed ulcers [14,15,16]. In addition, the CWIS showed sensitivity to healed wounds in a randomized clinical trial when different types of dressings were evaluated for DFU [16]. Furthermore, its domains were also strongly correlated with SF-36 subscales as the gold standard [8,15]. This suggests that CWIS is a valid disease-specific measure of QoL in subjects with DFU [11,15,17,18]. The CWIS, originally developed in English in the UK, has been validated in other countries, such as China, Sri Lanka, Canada, Sweden, and Portugal [15,17,18,19,20,21], while it has only been translated and culturally adapted in German, French, and US English [22]. In Spain, there is no available Spanish questionnaire to specifically assess the QoL of patients with chronic wounds [14]. Thus, the aim of the study was to translate the CWIS into Spanish and prospectively assess its validity and reliability in a group of patients with DFU.

## 2. Materials and Methods

### 2.1. Design and Settings

Participants were patients with DFU treated by an expert in chronic wounds at the Department of Endocrinology, University Hospital Arnau de Vilanova between June 2013 and January 2015. A description of the study participants has been provided in a previous publication [23]. The inclusion criteria were a diagnosis of diabetes mellitus (type 1 or type 2 diabetes) with a DFU; a first ulcer, or a new-onset ulcer with ≤3 months duration; over 18 years old; and the presence of one or more ulcers located below the malleoli. The exclusion criteria were having psychological or cognitive deterioration, having a terminal illness, or having been hospitalized. The ethics committee from the University Hospital Arnau de Vilanova approved the study. Written informed consent form was obtained from all participants.

### 2.2. Clinical and Sociodemographic Variables

A detailed description of clinical variables has been described in our previous publication [24]. These data were collected through individual interviews with all participants. Furthermore, a careful review of their clinical records was performed. Hypertension and dyslipidemia were determined if participants were specifically being treated with drugs for these two conditions. Diabetic foot disease and Charcot neuroarthropathy were diagnosed by performing a podiatric examination, as detailed in a previously published study [25]. After physical examination, a previous lower-limb amputations (minor or major), foot abnormalities, the presence of Charcot foot disease and an assessment of the local ulcer features was determined [25,26,27]. The diagnosis of DFU was established following the standard recommendations of the International Working Group on the Diabetic Foot (IWGDF) [28]. Peripheral arterial disease was appraised using the ankle-brachial index (ABI) and was categorized as normal (from 0.91 to 1.30), moderate ischemia (from 0.41 to 0.90), severe (from 0 to 0.40), and non-compressible because of the detection of calcification (more than 1.30) [29]. Moreover, the pedal or posterior tibial pulse was analyzed in those study participants with an ABI value over 1.30. The determination of peripheral arterial disease was defined by the presence of non-palpable pulses. Following the IWGDF consensus, the type of ulcer was classified as neuropathic, ischemic and neuroischemic [28]. The presence of two symptoms or greater of inflammation (i.e., redness, induration, warmth, and tenderness/pain), or purulent secretions was determined to diagnose an infection of ulcer. Moreover, signs of systemic inflammation (i.e., leukocytosis, fever and C reactive protein) were also evaluated and the grade of the infection was appropriately classified [30].

### 2.3. Instruments

The questionnaires (CWIS, SF-36 and EQ-5D) were administered at 7 days and 4, 12, and 26 weeks after the baseline assessment by individual interviews with each of the participants.

#### 2.3.1. Cardiff Wound Impact Schedule (CWIS)

CWIS was designed and validated to specifically assess the QoL of subjects with chronic wounds (leg ulcers and DFU) [15]. This questionnaire contains 47 items divided into four scales: demographic and clinical characteristics (3 items), global HRQoL (1 item), satisfaction with HRQoL (1 item) and impact of the wound on lifestyle. This last scale includes 3 domains: social life (14 items in total, 7 related to stress and 7 to experience), well-being (7 items), and physical symptoms and everyday living (24 items in total, 12 related to stress and 12 to experience). All three domains are scored on a 5-point scale, from “not at all” to “always”. The final score ranges from 0 (poorer QoL) to 100 points (higher QoL).

#### 2.3.2. 36-Item Short-Form Health Survey (SF-36)

The SF-36 questionnaire is a generic tool that evaluated the health status of the subject [14]. This contains 36 items that are grouped to eight subscales: physical role, physical functioning, general health, bodily pain, social functioning, vitality, emotional role, and mental health. These eight subscales are incorporated into physical and mental health summary scores. Each subscale ranged from 0 (poorer health status) to 100 points (better health status) and is normalized using US norms. This questionnaire is commonly used to validate other tools related to QoL.

#### 2.3.3. EuroQoL 5D Health Utility Index (EQ-5D)

This is a generic questionnaire designed used to assess HRQoL in different diseases, as well as in the general populations of several countries [9]. This instrument includes 5 dimensions: self-care, mobility, pain/discomfort, usual activities, and anxiety/depression. Each item is divided into three categories: no problems, some problems, and extreme problems. This questionnaire shows a visual analogue scale (VAS) to rate the current health status of the study participants on a scale scored from 0 (poorer health status) to 100 points (higher health status). An index value (EQ-5D index value) is calculated by combining the five dimensions using UK weights for health status defined by each combination.

### 2.4. Transcultural Adaptation of the CWIS Questionnaire

Two fluent translators in both languages translated the original English version independently to Spanish. The two translated versions were later compared by a group of experts and by a group of patients. Both groups discussed the differences between both versions and reached an agreement. A third translator back-translated the proposed Spanish version for the research group to compare this back-translation with the original English version and correct it if required. The content validity of the final Spanish version was assessed by seven experts using a Likert scale that ranged from 1 (of little relevance) to 4 points (very relevant). They evaluated the relevance of each item to assess the impact of DFU on the QoL of the patients (available at https://www.irblleida.org/media/upload/arxius/VARIS/Questionari_CWIS.pdf) (accessed on 2 September 2021).

### 2.5. Sample Size

The sample size was based on Cronbach’s alpha, a measure of internal consistency. Accepting an alpha risk of 0.05 and a beta risk of 0.2 in a two-sided test and without dropouts, 124 patients were needed to detect Cronbach’s alpha coefficients of 0.3 and higher as statistically significant. Anticipating a maximum dropout rate of 15%, the required minimum sample size was 143.

### 2.6. Statistical Analysis

Descriptive statistics, including mean and standard deviation for quantitative variables and absolute and relative frequencies for qualitative variables, were used.

The experts’ assessment for content validity was 3.63 on average (90% of the maximum score of 4). Reliability was measured by internal consistency and reproducibility. Internal consistency was measured using the α-Cronbach coefficient [31], where coefficients of 0.70 or higher were considered adequate in accordance with the study protocol. In addition, reproducibility or test-retest reliability was determined using the CWIS results at baseline and one day 7 visit after the first treatment for patients with no healed diabetic ulcers, assuming no changes for them. It was quantified using the intraclass correlation coefficient (ICC), defined by a single rater, two-way, mixed-effects model for quantitative variables.

Validity assessment was based on criterion and construct validity. In this study, the criterion validity was only determined in terms of concurrent validity with the domains and summary measures of the SF-36 and EQ-5D questionnaires by estimating Pearson’s correlation coefficients with the scores of the CWIS subscales. Criterion validity was considered for values over 0.30 (signifying moderate to large correlations). Construct validity was assessed using a confirmatory factor analysis of the three CWIS subscales for impact of the wound on lifestyle. The comparative fit index (CFI), the root mean square error of approximation (RMSEA), and the standardized root mean square residual (SRMR) were estimated. Values of 0.95 or higher, 0.06 or lower, and 0.08 or lower, respectively, are indicative of a good fit to the subscales of the original CWIS (i.e., the three subscales related to the impact of the wound on lifestyle).

The sensitivity to change over time was graphically assessed through the smoothed trends from baseline (visit 1) until visit 5 (visits corresponding to the questionnaire assessments at 7 days and at 4, 12, and 26 weeks or wound cure from baseline assessment) and depending on the healing state of the ulcer at the last available visit. Changes from their inclusion until the last available treatment visit between healed and non-healed patients were compared using the Mann–Whitney test. The R software [32] was used for statistical analysis, with a significance level of 0.05.

## 3. Results

The characteristics of the 141 participants recruited in the study are shown in Table 1. A high frequency of patients with type 2 diabetes (95.0%), hypertension (82.3%), dyslipidemia (61.7%), and neuropathy (92.9%) was observed in this sample. In addition, the study group showed a low educational level (40.4% had not even completed primary school). Macrovascular complications were observed in a high proportion of patients (89.4%). Neuropathic ulcer was the most prevalent etiology within the study group (61.7%).

The content validity had an average score of 3.63 (90.7% of the maximum score of 4). Internal consistency of the CWIS domains was acceptable, with a standardized Cronbach’s alpha of 0.715 for social life (items adding experience and stress), 0.729 for well-being, and 0.797 for physical symptoms and everyday living (items adding experience and stress) domains (Table 2). Internal consistency of the CWIS domains were not improved or only marginally improved by the deletion of items. In terms of reproducibility, the CWIS well-being domain showed moderate reproducibility (ICC = 0.63), while the other domains (i.e., social life, physical symptoms, HRQoL, and satisfaction with HRQoL) showed good reproducibility (ICC from 0.80 to 0.88).

The analysis of the concurrent criterion validity is shown in Table 3. Significant Pearson’s correlations were found between the impact of the wound on lifestyle domains and self-reported quality of life assessed by CWIS and the domains and summary scores of the SF-36 and EQ-5D. Thus, the CWIS social life domain was largely correlated with the SF-36 social functioning domain and moderately correlated with the SF-36 domains of role physical and overall physical component (*r* ≥ 0.4, *p* < 0.001), as well as with the SF-36 domains of physical functioning, vitality, bodily pain, emotional role, and mental component summary (*r* > 0.30, *p* < 0.001). The CWIS well-being assessment was moderately correlated with all EQ-5D and SF-36 domains except for SF-36 general health assessment. The CWIS symptoms assessment was largely correlated with the SF-36 domains of physical functioning, role and component summary, bodily pain, and social functioning (*r* ≥ 0.50, *p* < 0.001) and moderately correlated with the EQ-5D, for both the VAS and index, and with the SF-36 vitality and emotional role domains. The CWIS domains of HRQoL and satisfaction with HRQoL were largely correlated with mental health and mental component summary (*r* ≥ 0.50, *p* < 0.001) and moderately correlated with EQ-5D index and VAS and with SF-36 bodily pain, vitality, social functioning, and emotional role (*r* < 0.30, *p* < 0.001).

On the other hand, the confirmatory factor analysis assessing construct validity of the CWIS showed that their structural definition of domains was not validated (Figure 1). The CFI was only 0.69, the RMSEA was 0.09, and the SRMR was 0.10, indicating that the CWIS structure for subscales lacked construct validity. Exploratory factor analysis with three factors showed that items 3 (family overprotective) and 6 (not going out for fear of bumping wound) from the social life domain loaded more in the well-being domain, while items 3 (healing confidence) and 5 (wound unpleasant look) from the well-being domain loaded more in the social life domain. Among the items of the symptoms domain, items 5 (wound suppuration) and 8 (wound unpleasant smell) loaded more in the well-being domain, while items 10 (adapted footwear), 11 (amount of treatments) and 12 (economic cost) loaded more in the social life domain.

We analyzed whether the Spanish version of the CWIS had a high sensitivity to detect changes between healed and unhealed ulcers (Table 4). The three domains of the CWIS for impact of the wound on lifestyle (i.e., social life, well-being and physical symptoms) showed a high sensitivity to change according to the healed group (*p* < 0.001). The HRQoL and satisfaction with HRQoL domains did not show significant changes (*p* = 0.903 and *p* = 0.085, respectively).

## 4. Discussion

The psychometric validation of the Spanish version of the CWIS was acceptable in a sample of patients with DFU. Our results suggest that the internal consistency of the CWIS domains was acceptable, while the reproducibility was excellent in physical symptoms, global HRQoL, and satisfaction with HRQoL domains. The CWIS domains were strongly correlated with SF-36 and EQ-5D subscales demonstrating an excellent criterion validity of the instrument, although well-being was not correlated with these two generic questionnaires. However, the CWIS structure in Spanish was not validated, showing a poor construct validity. On the other hand, this instrument showed a high sensitivity to detect changes between healed and unhealed ulcers.

The results of this study suggest that the Spanish version of the CWIS showed good internal consistency. Nevertheless, other versions of the questionnaire have reported a higher internal consistency (i.e., higher Cronbach α coefficient than our Spanish version), including the Sri Lankan (Cronbach α = 0.89), Swedish (Cronbach α = 0.92), and Chinese (Cronbach α = 0.93) version, as well as the original English form (Cronbach α = 0.96) [15,18,19,20]. However, in our study, the well-being domain did not show a high Cronbach α coefficient. This was similar to the Sri Lankan and Swedish versions [19,20], whereas the Chinese and English versions found a high internal consistency for all the CWIS domains [15,18].

In the present study, the test-retest stability of the instrument showed a moderate reproducibility in the well-being domain. Moreover, reproducibility was good for social life, physical symptoms, global HRQoL, and satisfaction with HRQoL. This was discordant with the Sri Lankan questionnaire, which found a poor reproducibility for the well-being domain and acceptable results for the other CWIS domains [19]. However, our results were similar to the Swedish questionnaire as they showed an excellent stability for the physical domain, and an acceptable reproducibility for well-being and social life domains [20]. The original English CWIS also found a high level of reproducibility as well as the Canadian CWIS questionnaire [15,17].

Criterion validity of the CWIS was excellent with stronger correlations with EQ-5D and SF-36 subscales. This is aligned with the validation studies performed in the other countries. They found similar correlations between CWIS domains and SF-36 subscales, which has been extensively used to assess HRQoL [15,17,18,19,20]. This indicates that the Spanish CWIS is a valid disease-specific measure of QoL in patients with DFU, although the well-being domain was not well correlated with the EQ-5D and SF-36 subscales. This could be due to the fact that well-being is a measure that should be assessed with a specific questionnaire designed and validated for a specific purpose/disease area. Furthermore, generic instruments are designed to study health status or HRQoL and not for specific components of QoL like well-being [33].

The original English CWIS was designed and validated with a high construct validity [15]. This is in contrast with our results which showed a poorer construct validity due to the cultural differences between the countries and populations. In our study involving Spanish subjects, the lifestyle variables related with well-being, social life, and ulcer symptoms were differentially grouped. For this reason, some changes in these items of the questionnaire might improve the construct validity. In the Chinese version, authors had to delete one item due to the cultural setting of China [18]. However, a confirmatory factor analysis to determine the construct validity of the subsequently translated version of the questionnaire was not performed [17,19,20].

Our results showed a high sensitivity to detect changes between healed and unhealed ulcers in the social life, well-being, and physical domains. This is similar to the other translated versions of the CWIS, except for the original English questionnaire which did not find differences between both groups in any domain [15,17,18,19,20].

This study has some limitations. This sample of patients showed other comorbidities that could influence the results of the QoL. Moreover, reproducibility of the CWIS was performed at seven days from baseline, whereas patients with DFU were treated at baseline to ensure the ulcers were cared for correctly because these patients have a high-risk of complications and mortality. However, this study has several strengths. At this moment, this is the first study to assess the reliability and validity of a Spanish version of the CWIS questionnaire. Despite this being an instrument designed to assess HRQoL in patients with chronic wounds, this study reports good psychometric properties. Furthermore, the CWIS correlated with the SF-36 and EQ-5D measures, which confers more quality and precision in the validation process. Another strength is the prospective design of the study that can assess changes in QoL and ulcer status over time.

## 5. Conclusions

The Spanish version of the CWIS questionnaire showed an acceptable validity in some respects, such as reproducibility, criterion validity, sensitivity to ulcer changes over time, and reliability to assess the QoL of patients with DFU. However, its construct validity was poor, indicating cultural differences between populations from different countries. Therefore, we strongly feel that further studies in other Spanish settings are warranted.

## Figures and Tables

**Figure 1 jcm-10-04023-f001:**
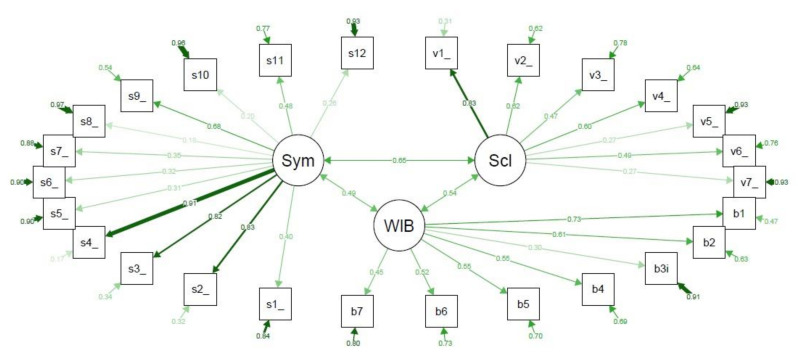
Confirmatory factor analysis of the Cardiff Wound Impact Schedule (CWIS) domains. Sym, Physical symptoms and everyday living; Scl, Social life; WIB, Well-being. v1–v7 are the items of the Social life domain. b1–b7 are the items of the Well-being domain. s1–s12 are the items of the Physical symptoms and everyday living domain. Each arrow between the questionnaire items and the subscale that they are measuring shows the standardized pattern coefficients for this relationship, where values closer to 1.0 (wider and darker) are indicative of better fit, and the circled arrow represented in each questionnaire item shows the residuals. The arrows connecting the subscales show the pairwise correlation between them. Comparative fit index (CFI) = 0.69; root mean square error of approximation (RMSEA) = 0.09; standardized root mean square residual (SRMR) = 0.10.

**Table 1 jcm-10-04023-t001:** Baseline clinical and sociodemographic characteristics of the study group.

Characteristics	Study Group (*n* = 141)
Age (years)	68.3 (13.3)
Sex (men)	95 (67.4)
Ethnicity (Caucasian)	140 (99.3)
Educational level	
Not even primary	57 (40.4)
Completed primary	47 (33.3)
Secondary high school	28 (19.9)
Graduate or higher	9 (6.4)
Employed	24 (17.0)
Smoking	
Never	63 (44.6)
Current or former	78 (55.4)
Type 2 diabetes	134 (95.0)
BMI, kg/m^2^	29.0 (4.9)
HbA1c, %	7.5 (1.6)
Hypertension	116 (82.3)
Dyslipidemia	87 (61.7)
Microvascular complications	
Retinopathy	96 (68.1)
Nephropathy	51 (36.2)
Neuropathy	131 (92.9)
Cardiovascular disease	126 (89.4)
Diabetes therapy	
OAD	41 (29.1)
OAD + insulin	57 (40.4)
Insulin	36 (25.5)
Diet	7 (5.0)
Antiplatelet agents	94 (66.7)
Dialysis	8 (5.7)
Type of ulcer	
Neuropathic	87 (61.7)
Ischemic	9 (6.4)
Neuroischemic	45 (31.9)
Infection of ulcer	83 (58.9)
Type of previous amputation	
Minor	41 (29.1)
Major	2 (1.4)
Presence of Charcot foot disease	9 (6.4)

Data are shown as mean (SD) for continuous variables or n (%) for categorical variables. BMI, body mass index; HbA1c, glycated hemoglobin.; OAD, oral antidiabetic drugs.

**Table 2 jcm-10-04023-t002:** Inter-item internal consistency and reproducibility of the Cardiff Wound Impact Schedule (CWIS) domains.

CWIS Domains	Number of Items ^1^	Range of Correlations ^2^	Average Inter-Item Correlation	Cronbach’s Alpha	Reproducibility ICC (95%CI)
Social life ^3^	14	−0.086–0.533	0.264	0.715	0.80 (0.72–0.85)
Well-being	7	0.019–0.565	0.278	0.729 ^5^	0.63 (0.51–0.73)
Physical symptoms and everyday living ^4^	24	0.040–0.767	0.247	0.797 ^6^	0.84 (0.79–0.88)
Self-reported HRQoL	1	-	-	-	0.87 (0.82–0.91)
Satisfaction with HRQoL	1	-		-	0.88 (0.83–0.91)

^1^ Number of items per domain. ^2^ Inter-item Pearson’s correlations. ^3,4^ Experience and stress item correlations are summated. ^5^ Improved from 0.729 to 0.743 if the third item for this subscale is deleted. ^6^ Improved from 0.797 to 0.800 if the eight item for this subscale is deleted. HRQoL, health-related quality of life; ICC, intraclass correlation coefficient; CI, confidence interval.

**Table 3 jcm-10-04023-t003:** Linear regression between domains of the Cardiff Wound Impact Schedule (CWIS) and the SF-36 and EQ-5D overall and subscale scores.

Domains	CWIS Domains				
	Social Life	Well-Being	Physical Symptoms and Everyday Living	Global HRQoL	Satisfaction with HRQoL
SF-36 Subscales					
Physical functioning	0.399 **	0.334 **	0.610 **	0.247 *	0.199 *
Role physical	0.443 **	0.356 **	0.528 **	0.228 *	0.251 *
Bodily pain	0.341 **	0.298 **	0.544 **	0.431 **	0.399 **
General health	0.212 *	0.240 *	0.250 *	0.348 **	0.390 **
Vitality	0.397 **	0.366 **	0.436 **	0.467 **	0.418 **
Social functioning	0.523 **	0.365 **	0.530 **	0.424 **	0.447 **
Role emotional	0.316 **	0.308 **	0.345 **	0.303 **	0.367 **
Mental health	0.284 *	0.330 **	0.281 *	0.511 **	0.524 **
Overall physical component ^1^	0.406 **	0.321 **	0.619 **	0.253 *	0.220 *
Overall mental component ^2^	0.327 **	0.336 **	0.268 *	0.493 **	0.535 **
EQ-5D subscales					
VAS	0.221 *	0.347 **	0.307 **	0.463 **	0.487 **
EQ-5D index value	0.261 *	0.315 **	0.419 **	0.396 **	0.365 **

^1,2^ calculated according to the SF-36 subscales involved physical and mental roles. HRQoL, health-related quality of life; VAS, visual analog scale. * *p* < 0.05; ** *p* < 0.001.

**Table 4 jcm-10-04023-t004:** Descriptive analysis of the sensitivity to change assessment of the Cardiff Wound Impact Schedule (CWIS) domains according to healing status.

Domains	Unhealed (*n* = 34)	Healed (*n* = 107)	*p*-Value
Change in Social life from baseline	2.7 (0.0–8.9)	12.5 (3.6–19.6)	<0.001
Change in Well-being from baseline	0.0 (−2.7–7.2)	35.7 (21.4–46.4)	<0.001
Change in Physical symptoms and everyday living	0.0 (0.0–10.4)	10.4 (4.2–16.7)	<0.001
Global HRQoL	0.0 (−1.0–1.0)	0.0 (−1.0–1.0)	0.903
Change in Satisfaction with HRQoL	0.0 (−1.0–0.0)	0.0 (−1.0–1.0)	0.085

Data are median (95% confidence interval). HRQoL, health-related quality of life.

## Data Availability

The data presented in this study are available on request from the corresponding author. The data are not publicly available due to privacy reasons.

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
