# Peer review of "Psychometric Validation of the Cardiff Wound Impact Schedule Questionnaire in a Spanish Population with Diabetic Foot Ulcer"

_jcm, 2021, doi:10.3390/jcm10174023_

Round 1
Reviewer 1 Report
important topic - thank you for the substantial work
the assessments described as 'baseline' may have also been of value during questionnaire administration points, as what if infection or change in ischaemic status occurred during study timeline? This could expose divergence between elements of the instruments used
line 177 requires close bracket then full stop before 'Construct validity'
would like more insight/suggestion (s) into why the cultural differences between UK and Spain translated into weaker construct validity and also how this could be modified to better suit the Spanish population
Author Response
RESPONSE TO THE REVIEWERS’ COMMENTS
Psychometric validation of the Cardiff Wound Impact Schedule questionnaire in a Spanish population with diabetic foot ulcer (Manuscript ID: jcm-1336704).
We highly appreciate the input given by the Reviewers, which enabled us to greatly improve the quality of our Manuscript. We are hereby enclosing our point-by-point responses to each of the Reviewer’s comments. Please, note that in the revised version of the Manuscript, changes are marked with the track changes tool, so that they can be easily traced.
REVIEWER 1
Comments and Suggestions for Authors
important topic - thank you for the substantial work
the assessments described as 'baseline' may have also been of value during questionnaire administration points, as what if infection or change in ischaemic status occurred during study timeline? This could expose divergence between elements of the instruments used
We greatly appreciate the Reviewer’s comments. The CWIS questionnaire is specifically designed and validated to assess QoL in subjects with ulcers; this instrument showed a higher sensitivity to discriminate between healed and unhealed wounds. In our study, we validated the sensitivity to ulcer changes of the instrument during the follow-up visits (Table 4). We showed that the CWIS questionnaire is highly sensitive to ulcer changes also considering both infection and ischemic changes. Please, note that this is explained in the paragraph on Clinical and sociodemographic variables (lines 102 – 104) of the Materials and Methods section.
line 177 requires close bracket then full stop before 'Construct validity'
We thank the Reviewer for this suggestion. We have amended this error in the revised version of the manuscript.
would like more insight/suggestion (s) into why the cultural differences between UK and Spain translated into weaker construct validity and also how this could be modified to better suit the Spanish population
In the Spanish population, the variables associated with lifestyle are differentially distributed and grouped in comparison with other countries due to the cultural differences between populations. The exploratory factor analysis of the CWIS showed that two items regarding social life had more weight on the well-being domain in our population. Moreover, two items of the well-being domain (healing confidence and wound unpleasant look) have a closer relationship with the social life domain in the Spanish population. Finally, some items of the symptoms’ domain are more closely associated with the well-being domain such as wound suppuration and wound unpleasant smell; however, the adapted footwear, number of treatments and economic costs have a higher weight on the social life domain. These results were already explained in the Results section (lines 297 - 312 of the revised version of the manuscript). Thus, as the Reviewer suggests, these changes in the questionnaire would improve the construct validity of the instrument in the Spanish population. For this reason, we have included a statement in the Discussion section of the manuscript, as follows:
Lines 370 - 373: In our study involving Spanish subjects, the lifestyle variables related with well-being, social life and ulcer symptoms were differentially grouped; for this reason, some changes in these items of the questionnaire might improve the construct validity.
Reviewer 2 Report
This is an interesting article that evaluated CWIS into Spanish patients and prospectively assess its validity and reliability in 141 patients with DFU. The article is well written, two minor points need clarification. The introduction should be expanded and it would be better to update the article with most recent references.
Author Response
RESPONSE TO THE REVIEWERS’ COMMENTS
Psychometric validation of the Cardiff Wound Impact Schedule questionnaire in a Spanish population with diabetic foot ulcer (Manuscript ID: jcm-1336704).
We highly appreciate the input given by the Reviewers, which enabled us to greatly improve the quality of our Manuscript. We are hereby enclosing our point-by-point responses to each of the Reviewer’s comments. Please, note that in the revised version of the Manuscript, changes are marked with the track changes tool, so that they can be easily traced.
REVIEWER 2
This is an interesting article that evaluated CWIS into Spanish patients and prospectively assess its validity and reliability in 141 patients with DFU. The article is well written, two minor points need clarification. The introduction should be expanded and it would be better to update the article with most recent references.
We greatly appreciate the Reviewer’s comments. Therefore, we have included some additional sentences in the Introduction section (see below). Moreover, we have updated the references list (from reference 3 to 13, and from reference 25 to 30 of the revised version). Please, see changes in the new version of the manuscript.
Lines 51 - 53: This condition is defined as an ulceration of the foot associated with diabetic neuropathy which show any grade of ischemia and infection [3].
Lines 56 – 61: The pathogenesis of DFU involves multiple factors such as peripheral neuropathy, artery disease, traumas and foot deformities, and abnormal joints [3]. Additionally, a recent meta-analysis about the global epidemiology of DFU demonstrated that subjects with DFU were older, had lower body mass index, longer diabetes duration, and showed higher frequency of hypertension, diabetic retinopathy and smoking habit in comparison with those without DFU [5].
Lines 68 – 72: Moreover, a poorer QoL is associated with several clinical factors such as pain, fatigue, wound infection, restricted mobility and social isolation [13]. Furthermore, QoL can be negatively influenced by other factors such as the frequency of attending clinic visits and hospitalizations, and the presence of disturbed daily life activities [13].